# Clinical, radiological and therapeutic characteristics of patients with COVID-19 in Saudi Arabia

**Mohammed Shabrawishi[1,2], Manal M. Al-Gethamy[3], Abdallah Y. Naser[4], Maher A. Ghazawi[5], Ghaidaa F. Alsharif[2], Elaf F. Obaid[2], Haitham A. Melebari[6], Dhaffer M. Alamri[3], Ahmad S. Brinji[5], Fawaz H. Al Jehani[7], Wail Almaimani[7], Rakan A. Ekram[8], Kasim H. Alkhatib[9], Hassan Alwafi**[10] *

1 Department of Respiratory Medicine, Al Noor Specialist Hospital, Mecca, Saudi Arabia, 2 Department of Internal Medicine, Al Noor Specialist Hospital, Mecca, Saudi Arabia, 3 Department of Infection Prevention and Control Programme, Al Noor Specialist Hospital, Mecca, Saudi Arabia, 4 Faculty of Pharmacy, Isra University, Amman, Jordan, 5 Department of Radiology, Al Noor Specialist Hospital, Mecca, Saudi Arabia, 6 Department of Internal Medicine, King Faisal General Hospital, Mecca, Saudi Arabia, 7 Al Noor Specialist Hospital, Mecca, Saudi Arabia, 8 School of Public Health and Health Informatics, Umm Al Qura University, Mecca, Saudi Arabia, 9 Intensive Care Department, Al Noor Specialist Hospital, Mecca, Saudi Arabia, 10 Faculty of Medicine, Umm Al Qura University, Mecca, Saudi Arabia

* hhwafi@uqu.edu.sa

**Data Availability Statement:** All relevant data are within the manuscript and its supporting information files.

## Abstract

### Background

Coronavirus disease 2019 (COVID-19) is a rapidly spreading global pandemic. The clinical characteristics of COVID-19 have been reported; however, there is limited research investigating the clinical characteristics of COVID-19 in the Middle East. This study aims to investigate the clinical, radiological and therapeutic characteristics of patients diagnosed with COVID19 in Saudi Arabia.

### Methods

This study is a retrospective single-centre case series study. We extracted data for patients who were admitted to the Al-Noor Specialist Hospital with a PCR confirming SARS-COV-2 between 12th and 31st of March 2020. Descriptive statistics were used to describe patients' characteristics. Continuous data were reported as mean ± SD. Chi-squared test/Fisher test were used as appropriate to compare proportions for categorical variables.

### Results

A total of 150 patients were hospitalised for COVID-19 during the study period. The mean age was 46.1 years (SD: 15.3 years). The most common comorbidities were hypertension (28.8%, n = 42) and diabetes mellitus (26.0%, n = 38). Regarding the severity of the hospitalised patients, 105 patients (70.0%) were mild, 29 (19.3%) were moderate, and 16 patients (10.7%) were severe or required ICU care.

**Funding:** The author(s) received no specific funding for this work.

**Competing interests:** The authors have declared that no competing interests exist.

## Conclusion

This case series provides clinical, radiological and therapeutic characteristics of hospitalised patients with confirmed COVID-19 in Saudi Arabia.

## Introduction

In early December 2019, a cluster of acute pneumonia of unknown aetiology was identified in Wuhan, China [1]. The pathogen was identified as a new RNA virus from the betacoronavirus family and was named "severe acute respiratory syndrome coronavirus 2" (SARS-CoV-2) [1]. The respiratory illness caused by the 2019 novel coronavirus disease (COVID-19) is highly infectious [2], and therefore, the World Health Organization (WHO) has characterized the diseases as a pandemic infection [3]. As of April 25, 2020, more than 2,700,000 confirmed cases were reported worldwide, and it has spread from Wuhan to more than 200 countries across the world [4].

The Kingdom of Saudi Arabia (KSA) is the largest country in the Arabian Peninsula and it is located in the south west part of Asia [5]. In a historical decision, KSA has suspended Umrah and all religious visits to the country in an attempt to prevent and delay the spread of COVID-19 in KSA. On March 2, 2020, Saudi Arabia confirmed its first case of COVID-19, which was imported from Iran [4]. Several other local clusters were identified later, with the majority of the cases being linked to recent travel history.

In recent studies, the clinical features and severity of COVID-19 have been described as being similar to other respiratory viruses such as severe acute respiratory syndrome (SARS) and Middle East respiratory syndrome (MERS) [6–8]. Symptoms can range from mild flu-like symptoms to acute respiratory distress syndrome (ARDS) [9]. However, the characteristics and the course of the disease in Middle Eastern populations remains unclear. Exploring the clinical characteristics of patients diagnosed with COVID-19 in Saudi Arabia is important, considering there are many visitors who travel to Saudi Arabia for religious purposes. Besides this, there is a high volume of air traffic for other purposes in this country, which was esti-mated to include around 39 million people in 2018. In 2019, around 7.5 million Muslim entered the holy city of Mecca for Umrah purposes [10]. This highlights how crucial it is to explore the characteristics of patients diagnosed with this infection, which is becoming increasingly widespread in the region. To address the aforementioned knowledge gaps, and given the ongoing spread of COVID-19 in the Middle East, this study aims to describe the clin-ical, radiological, and therapeutic characteristics of COVID-19 in a selected cohort of patients in Mecca, Saudi Arabia.

## Methods

### Study design and participants

This was a retrospective single-centre case series study of 150 patients diagnosed with COVID-19. We extracted data for patients who were admitted to Al-Noor Specialist Hospi-tal with a polymerase chain reaction (PCR) confirming SARS-COV-2 between 12th and 31st of March 2020. Al-Noor Specialist Hospital in Mecca, Saudi Arabia, is a 500-bed specialist and teaching hospital in the centre of the holy city of Mecca. It delivers tertiary care throughout the Mecca region of Saudi Arabia and it is part of the Ministry of Health services [11, 12]. All patients enrolled in this study were diagnosed with COVID-19 through real

time (RT)-PCR obtained through nasopharyngeal swabs, which were processed and validated through a regional lab. All data including outcomes, mortality and length of stay were monitored up to 8th April 2020.

### Data collection

Data were extracted from both paper and electronic records using a unique medical record number (MRN) for each patient. All data were reviewed and checked by a medical team, including two medical residents and a consultant pulmonologist. Extracted data included patients' demographics, comorbidities, history of recent travel and history of contact with a confirmed COVID19 patient in the past two weeks. In addition, clinical signs, symptoms, radiological findings and pharmacological treatment received were collected. The radiological examinations were interpreted by a certified consultant radiologist who was blinded from the clinical presentation of the patients. The severity assessment of the chest x-ray (CXR) was estimated subjectively. All data were collected at the time of the admission.

### Study variables

Data regarding the clinical progression and severity of the disease were reported as the worst classification reached at any point during hospitalisation. We further classified the severity of the disease based on the following criteria: 1) mild disease was defined as patients with upper respiratory tract symptoms (such as rhinorrhoea, sore throat, headache, myalgia, body pain, low-grade fever and /or dry cough) with the absence of clinical or radiological findings of pneumonia; 2) moderate disease was defined as symptomatic patients with radiological signs of pneumonia; 3) severe disease was defined as confirmed COVID-19 pneumonia with any of the following respiratory rates: $\geq$30/min, blood oxygen saturation $\leq$93% at rest, PaO2/FiO2 ratio <300, lung infiltration >50% of the lung field, and 4) critically severe disease was defined as any of the following: respiratory failure which required invasive mechanical ventilation, shock, or organ failure which required admission to the intensive care unit.

### Ethics

This study was approved by the institutional ethics board at the Ministry of Health in Saudi Arabia (No. H-02-K-076-0420-286). Patients were informed that their clinical data would be used for clinical or research purposes, while keeping all their personal information confidential. The need for informed consent was waived by the ethics committee.

### Statistical analysis

Descriptive statistics were used to describe patients' demographic characteristics, radiological findings, medication use, and comorbidities. Continuous data were reported as mean ± SD, and categorical data were reported as percentages (frequencies). Independent sample t test was used to compare the mean value for continuous variables. A Chi-squared test/Fisher test was used as appropriate to compare proportions for categorical variables. Logistic regression analysis was used to identify predictors of clinical characteristics. A confidence interval of 95% ($p < 0.05$) was applied to represent the statistical significance of the results and the level of significance was assigned as 5%. SPSS (Statistical Package for the Social Sciences) version 25.0 software (SPSS Inc.) was used to perform all statistical analysis.

## Results

### Patients' clinical characteristics

Table 1 presents patients' characteristics at presentation at the hospital. A total of 150 patients were hospitalised for COVID-19. The mean age was 46.1 years (SD: 15.3 years), and ranged between 11 and 87. Around 61.0% (n = 90) were males. Six patients (3.9%) reported working in the healthcare sector. The most common comorbidities were hypertension (28.8%, n = 42) and diabetes mellitus (DM) (26.0%, n = 38). The majority of the patients (56.0%; n = 84) were local residents. Around half of the patients (54.1%, n = 80) reported that they had a contact history with a traveller. In addition, the majority of the patients, 64.4% (n = 96), had a contact history with a COVID-19 patient. Regarding the severity of the hospitalised patients, 105 patients (70.0%) were mild, 29 (19.3%) were moderate, and 16 patients (10.7%) were severe or required ICU care. Of the 105 mild patients, around 31.3% (n = 47) were asymptomatic. Patients with comorbidities were more likely to have a severe outcome compared to other patients (p<0.05).

**Table 1. Patients demographic characteristics at presentation.**

| Demographics | All patients (n = 150) | Mild cases (n = 105) | Moderate cases (n = 29) | Severe/Intensive care unit cases (n = 16) | P-value |
|---|---|---|---|---|---|
| Age (years; mean (SD)) | 46.1 years (15.3) | 45.4 years (±16.0) | 46.7 years (±12.1) | 49.8 years (±15.7) | 0.550 |
| Gender | | | | | |
| Female No. (%) | 60 (40.0) | 47 (44.8) | 10 (34.5) | 3 (18.8) | 0.112 |
| Healthcare worker | | | | | |
| Yes No. (%) | 6 (4.0) | 6 (5.8) | 0 | 0 | 0.110 |
| Place of residency No. (%) | | | | | |
| Kingdom of Saudi Arabia | 84 (56.0) | 57 (54.3) | 15 (51.7) | 12 (75.0) | 0.084 |
| Other countries | 66 (44.0) | 48 (45.7) | 14 (48.3) | 4 (25.0) | 0.239 |
| Comorbidities No. (%) | | | | | |
| Hypertension | 42 (28.8) | 29 (27.6) | 10 (35.7) | 3 (23.1) | 0.627 |
| Diabetes mellitus | 38 (26.0) | 20 (19.0) | 11 (39.3) | 7 (53.8) | 0.005** |
| Coronary artery disease | 11 (7.5) | 5 (4.8) | 3 (10.7) | 3 (23.1) | 0.094 |
| Renal disease | 10 (6.8) | 5 (4.8) | 1 (3.6) | 4 (30.8) | 0.018* |
| Thyroid gland problem (hypothyroidism) | 9 | 4 (8.2) | 2 (11.1) | 3 (37.5) | 0.123 |
| Asthma | 4 (2.7) | 3 (2.9) | 0 | 1 (7.7) | 0.306 |
| Cancer | 2 (1.4) | 0 | 0 | 2 (15.4) | 0.007** |
| CVA | 1 (0.7) | 1 (1.0) | 0 | 0 | 0.718 |
| COPD | 1 (0.7) | 0 | 0 | 1 (7.7) | 0.086 |
| CLD | 1 (0.7) | 0 | 0 | 1 (7.7) | 0.086 |
| Tracing history No. (%) | | | | | |
| Recent travel history (Yes) No. (%) | 65 (43.9) | 47 (45.6) | 14 (48.3) | 4 (25.0) | 0.263 |
| Contact with traveller (Yes) No. (%) | 80 (54.1) | 57 (55.3) | 18 (62.1) | 5 (31.3) | 0.124 |
| Contact with COVID-19 patient (Yes) No. (%) | 96 (64.4) | 71 (68.3) | 20 (69.0) | 5 (31.3) | 0.013* |
| Outcome (n = 148) No. (%) | | | | | |
| Deceased | 4 (2.7) | 2 (1.9) | 0 | 2 (12.5) | 0.086 |
| Improved | 47 (31.8) | 37 (35.2) | 6 (22.2) | 4 (25.0) | 0.358 |
| Not recovered | 3 (2.0) | 2 (1.9) | 1 (3.7) | 0 | 0.615 |
| Recovered | 94 (63.5) | 64 (61.0) | 20 (74.1) | 10 (62.5) | 0.434 |

Abbreviations; COVID-19: coronavirus disease-2019; CVA: cerebrovascular accident; COPD: chronic obstructive pulmonary disease; CLD: chronic liver disease; SD: Standard deviation; No: Number (frequency)

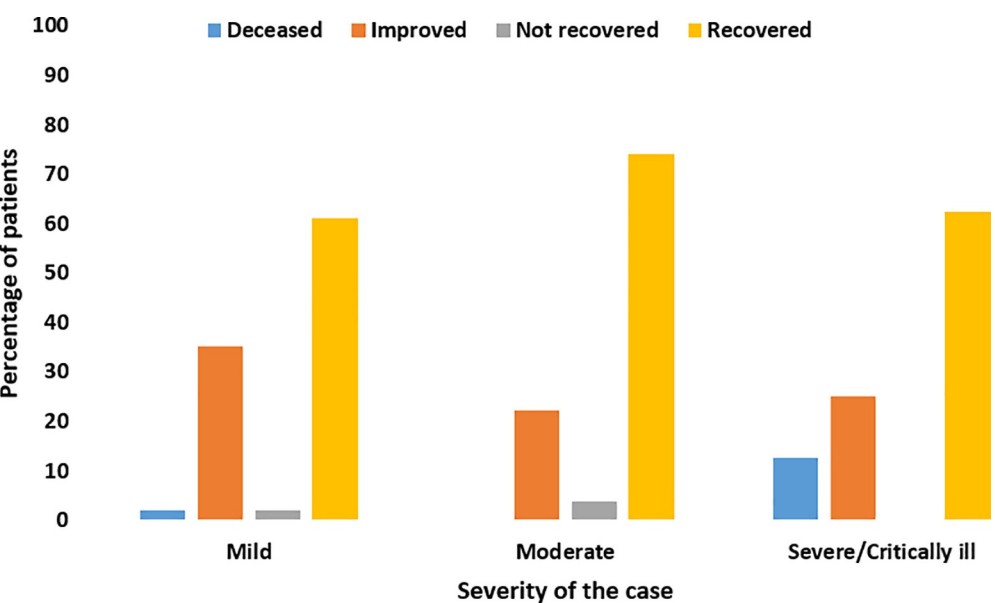

**Fig 1. Clinical severity stratified by gender.**

Patients who reported a contact history with a COVID-19 patient were more likely to have mild to moderate severity of the disease (p<0.05). Mild cases were more prevalent among females, while moderate to severe and or critical were prevalent among males (Fig 1).

For symptomatic patients, the most common symptoms at presentation were fever (49.3%, n = 72), dry cough (48.6%, n = 71), and shortness of breath (19.9%, n = 29) (Table 2). Furthermore, during admission, fever and cough (28%) were the most common symptoms followed by nausea and vomiting (12%). Most of the asymptomatic patients were females (OR: 0.45 [95%CI 0.22–0.92]; p = 0.027). In addition, patients who reported travel history or contact

**Table 2. Patient signs and symptoms at presentation and during admission.**

| Variable | Symptoms | | P-vale |
|---|---|---|---|
| | At presentation No. (%) | During admission No. (%) | |
| Fever | 72 (49.3) | 28 (19.2) | 0.029* |
| Cough | 71 (48.6) | 28 (19.2) | 0.024* |
| Shortness of breath | 29 (19.9) | 7 (4.8) | 0.000*** |
| Sore throat | 24 (16.4) | 2 (1.4) | 0.269 |
| Runny nose | 9 (6.2) | 0 (0.0) | >0.99 |
| Sputum | 5 (3.4) | 1 (0.7) | 0.034* |
| Headache | 4 (2.7) | 0 (0.0) | >0.99 |
| Myalgia | 4 (2.7) | 1 (0.7) | 0.813 |
| Diarrhea | 2 (1.4) | 5 (3.4) | 0.068 |
| Nausea/vomiting | 1 (0.4) | 12 (8.2) | 0.678 |
| Haemoptysis | 1 (0.4) | 1 (0.7) | 0.887 |
| Fatigue | 1 (0.4) | 1 (0.7) | 0.907 |

* p<0.05

**p<0.01

***p<0.000

with a traveller recently were three times (OR: 3.13 [95%CI 1.52–6.45]; p = 0.002) and four times (OR: 4.03 [95%CI 1.84–8.81]; p = 0.000) more likely to be asymptomatic, respectively. Besides, patients who reported contact with COVID-19 patients were four times more likely to be symptomatic (OR: 4.50 [95%CI 1.84–10.99]; p = 0.001).

## Radiological findings

Around half of the patients (49.7%, n = 72) had a normal radiological exam at presentation. The severity of the cases was correlated with an increase in the prevalence of GGO at presentation (P = 0.002). The predominant pattern of abnormality observed was ground-glass opacification (29.0%, n = 42), peripheral (57.5%, n = 42), and bilateral (35.3%, n = 35), which mainly involved the lower lobes (Fig 2). Most of the patients had stable radiological exams on follow up. Around 64.6% (n = 62) showed progression, half of them belonging to the more severe group (Table 3).

## Recovery

Patients stayed at the hospital for a mean duration of 9.2 days (SD: 3.9). The duration of stay in hospital ranged from two days to 23 days. At the end of the follow-up period, a total of 94 patients (63.5%) recovered and 31.8% (n = 47) improved clinically, but their RT-PCR results were still positive. On the other hand, three patients (2.0%) did not fully recover and four patients (2.7%) deceased. There was no statistically significant difference based on age regarding the recovery or whether the patient was symptomatic or asymptomatic upon presentation at the hospital (p>0.05). The majority of the patients with mild cases improved or recovered; however, there was no statistically significant difference between cases of different severity and recovery rate (p>0.05) (Fig 3).

## Therapeutic management

Beside supportive care, there were three main types of therapies that were prescribed to the patients for the management of COVID-19, including: a) antiviral therapy, b) antibiotics, and c) antimalarial medications (Table 4). A total of 6 patients (4.0%) received the three classes of treatment on the first day of their admission. The most commonly used antibiotics were macrolide monotherapy (12.7%, n = 19) (azithromycin or clarithromycin), followed by macrolide and cephalosporin combination therapy (8.7%, n = 13).

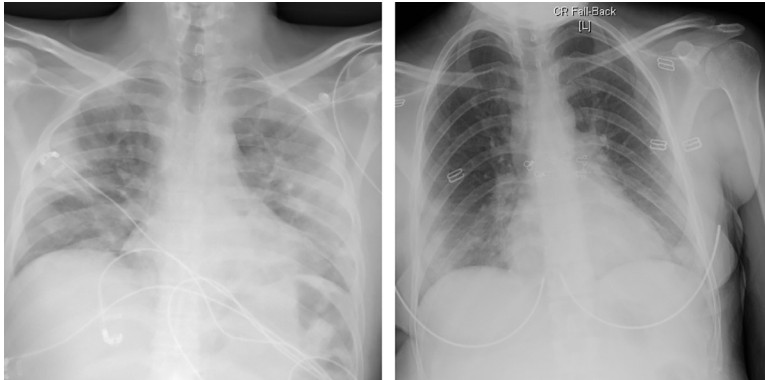

**Fig 2. CXR's of two different patients showing the most common abnormalities: Bilateral, peripheral ground glass opacities and consolidation.**

**Table 3. Radiological findings.**

| | Radiological findings (CXR) upon admission | | | | |
|---|---|---|---|---|---|
| | All patients (n = 150) | Mild cases (n = 105) | Moderate cases (n = 29) | Severe/Intensive care unit cases (n = 16) | P-value |
| **Predominant finding** | | | | | |
| Normal | 72 (49.7) | 62 (60.2) | 7 (25.9) | 3 (20.0) | 0.000*** |
| Ground glass opacity | 42 (29.0) | 21 (20.4) | 13 (48.1) | 8 (53.3) | 0.002* |
| Consolidation | 26 (17.9) | 16 (15.5) | 6 (22.2) | 4 (26.7) | 0.488 |
| Linear atelectasis | 3 (2.1) | 3 (2.9) | 0 | 0 | 0.354 |
| Diffusion reticular opacities | 1 (0.7) | 1 (1.0) | 0 | 0 | 0.795 |
| Reticulation | 1 (0.7) | 0 | 1 | 0 | 0.183 |
| **Distribution within the lobe** | | | | | |
| Central | 10 (13.7) | 6 (14.6) | 2 (10.0) | 2 (16.7) | 0.833 |
| Diffuse | 21 (28.8) | 14 (34.1) | 4 (20.0) | 3 (25.0) | 0.494 |
| Peripheral | 42 (57.5) | 21 (51.2) | 14 (70.0) | 7 (58.3) | 0.378 |
| **Distribution within the lung** | | | | | |
| Lower | 24 (32.9) | 12 (29.3) | 6 (30.0) | 6 (50.0) | 0.385 |
| Lower middle | 22 (30.1) | 10 (24.4) | 9 (45.0) | 3 (25.0) | 0.236 |
| Lower and middle and upper | 10 (13.7) | 8 (19.5) | 2 (10.0) | 0 | 0.089 |
| Diffuse | 10 (13.7) | 6 (14.6) | 2 (10.0) | 2 (16.7) | 0.833 |
| Peripheral | 2 (2.7) | 2 (4.9) | 0 | 0 | 0.309 |
| Middle | 2 (2.7) | 2 (4.9) | 0 | 0 | 0.309 |
| Upper | 1 (1.4) | 1 (2.4) | 0 | 0 | 0.559 |
| Upper and middle | 1 (1.4) | 0 | 0 | 1 (8.3) | 0.159 |
| No zonal predominance | 1 (1.4) | 0 | 1 (5.0) | 0 | 0.269 |
| **Laterality** | | | | | |
| Bilateral | 53 (35.3) | 27 (25.7) | 16 (55.2) | 10 (62.5) | 0.000*** |
| Unilateral right | 12 (16.4) | 10 (24.4) | 2 (10.0) | 0 | 0.035* |
| Unilateral left | 8 (11.0) | 4 (9.8) | 2 (10.0) | 2 (16.7) | 0.805 |
| **Progression** | | | | | |
| Stable | 62 (64.6) | 49 (74.2) | 11 (57.9) | 2 (18.2) | 0.001** |
| Worsen | 34 (35.4) | 17 (25.8) | 8 (42.1) | 9 (81.8) | |

CXR: chest x-ray

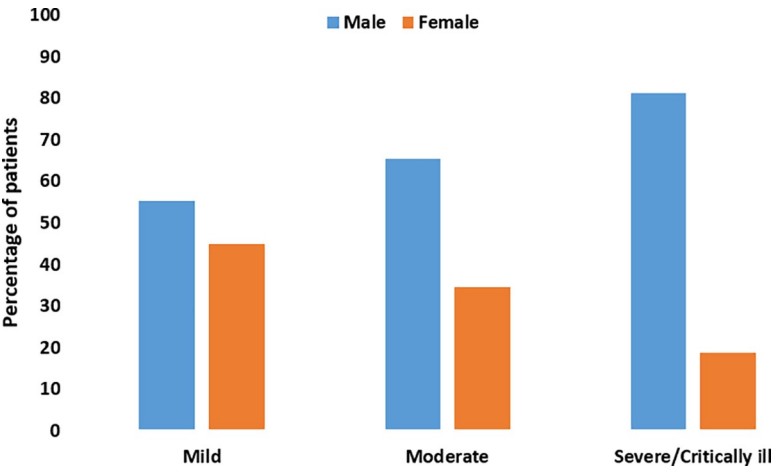

**Fig 3. Recovery rates stratified by case severity.**

**Table 4. Initial treatment characteristics.**

| Treatment therapy | Frequency (%) |
|---|---|
| Antiviral therapy | |
| Combination of antiretroviral (lopinavir and ritonavir) and ribavirin | 14 (9.3) |
| Antimalarial therapy | |
| Hydroxychloroquine | 25 (16.7) |
| Chloroquine | 15 (10.0) |
| Antibiotics therapy | 58 (38.7) |

## Discussion

To the best of our knowledge, this is the first and largest study to examine the clinical, radiological and therapeutic characteristics of COVID-19 in the Middle East region. We investigated clinical, radiological, and therapeutic characteristics of COVID-19 in 150 hospitalised patients in Saudi Arabia. We found that around 89.0% of the cases were either mild or moderate and only 11.0% were either severe or critical. Our finding showed that the clinical severity of COVID-19 was of a milder presentation compared to results from China [13], Italy [14] and the United States [15, 16]. These findings could be attributed to several factors including age and other demographic differences. The mean age in our study was 46.1 years (SD: 15.3), which was younger than the age reported in other studies. Several studies have reported poorer outcomes among older populations and patients with COVID-19 and comorbidities [17–19]. However, it is difficult to draw a causal inference and we urge for further studies to investigate this association. In addition, it is important to highlight that the majority of the Saudi Arabian population are younger than 44 years [20].

Male patients with COVID-19 were more prevalent in our study compared to females; this was also similar to previous reports which highlight that more males are infected with COVID-19 [1, 15]. These numbers could be attributed to sex-based immunological differences, or they could also be because of behavioural patterns such as smoking [21]. In addition, comorbidities are more prevalent in men, which could also be a reason for this difference [22]. However, there is a need for more research which focuses on gender differences and clinical outcomes with COVID-19.

Our study highlighted that around 28.8% and 26.0% of the study population had hypertension (HTN) and DM; these results were similar to previous reports that investigated the clinical characteristics of COVID-19 (1). Patients with DM and hypertension have an increased risk of complications of COVID-19, including acute respiratory distress syndrome (ARDS) [23]; however, the mechanism of this remains un-investigated and it is unclear whether patients with uncontrolled blood pressure have poorer outcomes of COVID-19 compared to patients with controlled blood pressure. In addition, angiotensin-converting enzyme (ACE) inhibitors and angiotensin receptor blockers (ARBs) are two commonly prescribed medications for the management of HTN, and since SARS-CoV-2 binds to ACE2 in the lung, some theories have been raised about the benefits of these medications in the treatment of COVID-19 [24].

SARS-COV2 has been described to be similar to seasonal influenza, SARS-COV and MERS; this includes the fact that it is transmitted through respiratory droplets [25, 26]. In addition, SARS-COV2 has similar symptoms to SARS-COV and MERS, such as fever, cough, and shortness of breath. This was reported in our study and it was also in line with previous studies [6, 14]; however, SARS-COV2 has a higher case fatality rate in comparison to seasonal flu (0.1%), while it is also milder in comparison to other respiratory viruses, such as SARS-COV (9.5%) and MERS (34.3%) [27]. Besides this, COVID-19 is a highly infectious pathogen

[28, 29], with some reports suggesting that half of the population of the United Kingdom (UK) has been infected without showing any symptoms or with having a mild course of the disease [30]. Our study demonstrated that around 31.3% of the study sample were asymptomatic and had mild disease. Mostly, these patients were identified through contact tracing and were isolated in the earlier course of the disease; whether this approach has any impact on the clinical course psychologically might need to be addressed in future studies. In addition, the majority of these patients had had contact with a confirmed COVID-19 patient, which may raise concerns regarding the mechanism and the underlying inflammatory response in these patients. More research is encouraged to investigate the characteristics of asymptomatic patients and if early detection and supportive treatment have a role in the clinical progression of the disease.

In our study, and unlike previous reports, nearly half of the patients presented with normal CXR; most of them were asymptomatic or had a mild disease. Furthermore, normal CXR at presentation may have a prognostic rule as only a few of those patients progressed into more severe cases. On the other hand, the presence of ground glass opacity is linked with a more aggressive course. The patterns found in abnormal exams were similar to the previously published reports and findings where peripheral, bilateral ground glass opacification was observed [31].

Our study highlighted that around 26.7% of the patients received antimalarial treatment and around 9.0% received antiviral treatment. These medications have been suggested to have some beneficial effect to reduce the viral load and eliminate the disease; however, there are also uncertainties regarding their safety [32, 33]. In addition, there has been debate about their efficacy in the treatment of COVID-19, with several trials now in the pipeline for the testing of these medications [34]. To date, there is no treatment for COVID-19, and the main approach in the management of the disease is to provide supportive treatment and to control the symptoms, including with the use of mechanical ventilators for critical cases [35].

This study provides some important messages, including similarities with previous reports from other countries about the clinical picture of COVID-19. It also highlights the low mortality rate which may reflect the early response of the Saudi Government and the good care provided for people living in the kingdom.

This study has some limitations. First, the number of patients included in the study was small. Second, the study population only included patients from a single-centre hospital in Saudi Arabia.

## Conclusion

This case series provides clinical, radiological, and therapeutic characteristics of hospitalised patients with confirmed COVID-19 in Saudi Arabia. Our study demonstrates similar characteristics of COVID-19 to previously reported studies worldwide.

## Author Contributions

**Conceptualization:** Mohammed Shabrawishi, Wail Almaimani, Hassan Alwafi.

**Data curation:** Mohammed Shabrawishi, Abdallah Y. Naser, Maher A. Ghazawi, Ghaidaa F. Alsharif, Elaf F. Obaid, Dhaffer M. Alamri, Hassan Alwafi.

**Formal analysis:** Abdallah Y. Naser, Hassan Alwafi.

**Investigation:** Mohammed Shabrawishi, Maher A. Ghazawi, Ghaidaa F. Alsharif, Elaf F. Obaid, Haitham A. Melebari.

**Methodology:** Mohammed Shabrawishi, Abdallah Y. Naser, Hassan Alwafi.

**Resources:** Mohammed Shabrawishi, Manal M. Al-Gethamy, Maher A. Ghazawi, Ghaidaa F. Alsharif, Elaf F. Obaid, Haitham A. Melebari, Dhaffer M. Alamri, Ahmad S. Brinji, Fawaz H. Al Jehani, Wail Almaimani, Rakan A. Ekram, Kasim H. Alkhatib.

**Supervision:** Mohammed Shabrawishi, Hassan Alwafi.

**Validation:** Mohammed Shabrawishi, Abdallah Y. Naser, Maher A. Ghazawi, Ahmad S. Brinji, Hassan Alwafi.

**Visualization:** Mohammed Shabrawishi, Hassan Alwafi.

**Writing – original draft:** Mohammed Shabrawishi, Abdallah Y. Naser, Maher A. Ghazawi, Hassan Alwafi.

**Writing – review & editing:** Mohammed Shabrawishi, Manal M. Al-Gethamy, Abdallah Y. Naser, Ghaidaa F. Alsharif, Elaf F. Obaid, Haitham A. Melebari, Dhaffer M. Alamri, Ahmad S. Brinji, Fawaz H. Al Jehani, Wail Almaimani, Rakan A. Ekram, Kasim H. Alkhatib, Hassan Alwafi.

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
