## [Decision Letter · Decision Letter 0]

8 Jul 2020

PONE-D-20-17611

Clinical, Radiological and Therapeutic Characteristics of Patients with COVID-19 in Saudi Arabia

PLOS ONE

Dear Dr. Alwafi,

Thank you for submitting your manuscript to PLOS ONE. After careful consideration, we feel that it has merit but does not fully meet PLOS ONE’s publication criteria as it currently stands. Therefore, we invite you to submit a revised version of the manuscript that addresses the points raised during the review process.

We look forward to receiving your revised manuscript.

Kind regards,

Wen-Jun Tu

Academic Editor

PLOS ONE

Journal Requirements:

2) In the ethics statement in the manuscript and in the online submission form, please provide additional information about the patient records used in your retrospective study. Specifically, please ensure that you have discussed whether all data were fully anonymized before you accessed them and/or whether the IRB or ethics committee waived the requirement for informed consent. If patients provided informed written consent to have data from their medical records used in research, please include this information.

3)  We suggest you thoroughly copyedit your manuscript for language usage, spelling, and grammar. If you do not know anyone who can help you do this, you may wish to consider employing a professional scientific editing service.  

4)  We note that you have indicated that data from this study are available upon request. PLOS only allows data to be available upon request if there are legal or ethical restrictions on sharing data publicly. For information on unacceptable data access restrictions, please see http://journals.plos.org/plosone/s/data-availability#loc-unacceptable-data-access-restrictions.

5) PLOS requires an ORCID iD for the corresponding author in Editorial Manager on papers submitted after December 6th, 2016. Please ensure that you have an ORCID iD and that it is validated in Editorial Manager. To do this, go to ‘Update my Information’ (in the upper left-hand corner of the main menu), and click on the Fetch/Validate link next to the ORCID field. This will take you to the ORCID site and allow you to create a new iD or authenticate a pre-existing iD in Editorial Manager. Please see the following video for instructions on linking an ORCID iD to your Editorial Manager account: https://www.youtube.com/watch?v=_xcclfuvtxQ

Reviewers' comments:

Reviewer's Responses to Questions

**Comments to the Author**

1. Is the manuscript technically sound, and do the data support the conclusions?

Reviewer #1: Yes

Reviewer #2: Yes

2. Has the statistical analysis been performed appropriately and rigorously? 

Reviewer #1: Yes

Reviewer #2: Yes

3. Have the authors made all data underlying the findings in their manuscript fully available?

Reviewer #1: Yes

Reviewer #2: Yes

4. Is the manuscript presented in an intelligible fashion and written in standard English?

Reviewer #1: Yes

Reviewer #2: Yes

5. Review Comments to the Author

Reviewer #1: This is a descriptive study of 150 patients with COVID-19 in Mecca city. The importance of this study arises from the fact that it presents data from Mecca, a city that is visited by more than 10 million people from all over the world yearly.

Additionally, it is the first study to report data about COVID-19 patients from the Middle East Region.

I am not sure if the authors have made all data underlying the findings in their manuscript fully available.

Reviewer #2: Coronavirus disease 2019 (COVID-19) is a rapidly spreading global pandemic. The clinical characteristics of COVID-19 has been reported; however, there are limited researches that investigated the clinical characteristics of COVID-19 in the Middle East. The aim of this study is to investigate the clinical, radiological and therapeutic characteristics of patients diagnosed with COVID19 in Saudi Arabia. This case series provides clinical, radiological and therapeutic characteristics of hospitalised patients with confirmed COVID-19 in Saudi Arabia.

Comments:

1. Please provide some additional detail in the text about the antiviral, antibiotic, glucocorticoid and Chinese traditional therapies received by the patients. What were the most common regimens used, what was the timing of initiation of antiviral therapy relative to onset of symptoms, what proportion of patients received all three classes of treatment, and other relevant details.

2. The comments about natural history should be tempered further, especially acknowledging potential confounders.

3. Please define how patients were selected for inclusion in this analysis. Were there others the authors did not choose to include and if so how did they differ from the patients selected? How was laboratory confirmation of these cases achieved? Were samples sent to a central laboratory? What assays were used to confirm the cases?

4. The possible clinical perspective should be added

5. The following references should be indexed in the revision text.

Petropoulos, F., & Makridakis, S. (2020). Forecasting the novel coronavirus COVID-19. PloS one, 15(3), e0231236.

Clinical Features and Short-term Outcomes of 102 Patients with Corona Virus Disease 2019 in Wuhan, China. Clinical Infectious Diseases, DOI: 10.1093/cid/ciaa243/5814897.

6. PLOS authors have the option to publish the peer review history of their article (what does this mean?). If published, this will include your full peer review and any attached files.

Reviewer #1: **Yes: **Ahmed S. BaHammam

Reviewer #2: No

---

## [Author Response · Author response to Decision Letter 0]

19 Jul 2020

Manuscript ID; PONE-D-20-17611

Title: Clinical, Radiological and Therapeutic Characteristics of Patients with COVID-19 in Saudi Arabia

Corresponding Author: Dr. Hassan Alwafi

Dear Editor, 

Thank you for the opportunity to revise and resubmit our manuscript based on the reviewers’ comments. Please find below our itemized point-by-point responses to the journal requirements and reviewers’ comments. Answers are written in blue font and edited text has been highlighted with track changes in the marked version of the manuscript.

Journal Requirements:

Thank you for your comment. We have addressed this comment.

2) In the ethics statement in the manuscript and in the online submission form, please provide additional information about the patient records used in your retrospective study. Specifically, please ensure that you have discussed whether all data were fully anonymized before you accessed them and/or whether the IRB or ethics committee waived the requirement for informed consent. If patients provided informed written consent to have data from their medical records used in research, please include this information.

Thank you for your comment. This study was approved by the institutional ethics board at the Ministry of Health in Saudi Arabia (No. H-02-K-076-0420-286). Patients were informed that their clinical data will be used for clinical or research purposes with keeping all their personal information confidential. The need for informed consent was waived by the ethics committee. We have now added this in the methods section in the main manuscript, please see lines (123 and 124).

3) We suggest you thoroughly copyedit your manuscript for language usage, spelling, and grammar. If you do not know anyone who can help you do this, you may wish to consider employing a professional scientific editing service.

Thank you for your suggestion. We have proofread the manuscript.

London Proofreaders 

Details as follow; 

Pop Brixton, 49 Brixton Station Rd, Brixton, London SW9 8PQ, United Kingdom

• A copy of your manuscript showing your changes by either highlighting them or using track changes (uploaded as a *supporting information* file).

• 4) We note that you have indicated that data from this study are available upon request. PLOS only allows data to be available upon request if there are legal or ethical restrictions on sharing data publicly. For information on unacceptable data access restrictions, please see http://journals.plos.org/plosone/s/data-availability#loc-unacceptable-data-access-restrictions.

All relevant data are within the manuscript and its supporting information files 

• 5) PLOS requires an ORCID iD for the corresponding author in Editorial Manager on papers submitted after December 6th, 2016. Please ensure that you have an ORCID iD and that it is validated in Editorial Manager. To do this, go to ‘Update my Information’ (in the upper left-hand corner of the main menu), and click on the Fetch/Validate link next to the ORCID field. This will take you to the ORCID site and allow you to create a new iD or authenticate a pre-existing iD in Editorial Manager. Please see the following video for instructions on linking an ORCID iD to your Editorial Manager account: https://www.youtube.com/watch?v=_xcclfuvtxQ

We have now added the ORCID ID of the corresponding author in the online system.

PONE-D-20-17611

Clinical, Radiological and Therapeutic Characteristics of Patients with COVID-19 in Saudi Arabia

Reviewer #1: This is a descriptive study of 150 patients with COVID-19 in Mecca city. The importance of this study arises from the fact that it presents data from Mecca, a city that is visited by more than 10 million people from all over the world yearly.

Additionally, it is the first study to report data about COVID-19 patients from the Middle East Region.

I am not sure if the authors have made all data underlying the findings in their manuscript fully available.

Thank you for your constructive comment. We agree with the author that Mecca city is visited by a large number of people every year for religious purposes. In addition, it is the third largest city population in Saudi Arabia (1). However, our study describe hospitalised data from a single centre (Al-Noor hospital) and we did not have access to any other hospital or nationwide data. Our data were collected from 12th of March until 31 of March, while the first case of COVID-19 reported in Saudi Arabia was in 2nd of March 2020 (2). In addition, during the early phases of the pandemic, the Saudi government were requesting patients who test positive for COVID-19 and who are stable to be quarantined in hotels and not to be admitted to hospitals. Therefore, it is reasonable to assume the small number of patients included in the study. 

Reviewer #2: Coronavirus disease 2019 (COVID-19) is a rapidly spreading global pandemic. The clinical characteristics of COVID-19 has been reported; however, there are limited researches that investigated the clinical characteristics of COVID-19 in the Middle East. The aim of this study is to investigate the clinical, radiological and therapeutic characteristics of patients diagnosed with COVID19 in Saudi Arabia. This case series provides clinical, radiological and therapeutic characteristics of hospitalised patients with confirmed COVID-19 in Saudi Arabia.

Comments:

1. Please provide some additional detail in the text about the antiviral, antibiotic, glucocorticoid and Chinese traditional therapies received by the patients. What were the most common regimens used, what was the timing of initiation of antiviral therapy relative to onset of symptoms, what proportion of patients received all three classes of treatment, and other relevant details.

Thank you for your suggestion. We have now addressed this comment, please see lines (189 - 192)

Regarding initiation of antiviral. Our aim in the study was to describe and endorse the different therapeutic options that were used in treating the patients and these were according to the Saudi Ministry of Health (MOH) protocol during the time of data collection. Patients were given antiviral treatment if they had COVID-19 symptoms for less than 10 days. 

2. The comments about natural history should be tempered further, especially acknowledging potential confounders.

Thank you for your comment. We agree with the author that more details about the history of patients is important. However, we did not have access to further details except those in the medical records of the patients. In addition, it is important to highlight that due to the nature of this descriptive study we assume that any missing patient’s history data will have a significant impact of the conclusion of this study, as there are no correlation of association being measured. 

3. Please define how patients were selected for inclusion in this analysis. Were there others the authors did not choose to include and if so how did they differ from the patients selected? How was laboratory confirmation of these cases achieved? Were samples sent to a central laboratory? What assays were used to confirm the cases?

Thank you for your comment. All patients who were hospitalized between 12th of March and 31st of March and tested positive for COVID-19 through RT-PCR nasopharyngeal swabs were included in this study. The laboratory tests for COVID-19 were processed and validated in the regional lab. We have now added this in the methods section in the main manuscript, please see lines (95 and 96).

4. The possible clinical perspective should be added

Thank you for suggestion. This study provides some important messages, this includes a similarity with previous reports from other countries about the clinical picture of COVID-19. It also highlights the low mortality rate which may reflect the early response of the Saudi Government and the good care provided for people living in the kingdom. We have added this in the discussion section in the main manuscript, please see lines (253 - 256).

5. The following references should be indexed in the revision text.

Petropoulos, F., & Makridakis, S. (2020). Forecasting the novel coronavirus COVID-19. PloS one, 15(3), e0231236.

Clinical Features and Short-term Outcomes of 102 Patients with Corona Virus Disease 2019 in Wuhan, China. Clinical Infectious Diseases, DOI: 10.1093/cid/ciaa243/5814897.

We have now added the above mentioned references.

References:

1. Statistics GAf. Statistical Yearbook of 2016 2016 [Available from: https://www.stats.gov.sa/en/5305.

2. MOH. COVID 19 Dashboard: Saudi Arabia 2020 [Available from: https://covid19.moh.gov.sa/.

---

## [Editor Report · Decision Letter 1]

22 Jul 2020

Clinical, Radiological and Therapeutic Characteristics of Patients with COVID-19 in Saudi Arabia

PONE-D-20-17611R1

Dear Dr. Alwafi,

We’re pleased to inform you that your manuscript has been judged scientifically suitable for publication and will be formally accepted for publication once it meets all outstanding technical requirements.

Kind regards,

Wen-Jun Tu

Academic Editor

PLOS ONE